# The Impact of Clothing E-Store Image on Intention Based on Search and Purchase Phases: From the Perspective of Sustainable Marketing

**Liyuan Jiang, Quanxi Li \* and Xiaoding Wu**

School of Business and Management, Jilin University, Changchun 130012, China
\* Correspondence: qxli@jlu.edu.cn

**Abstract:** With the continuous emergence of global warming and excessive waste of resources, especially in the clothing industry, sustainable clothing has become a fast-moving consumer goods trend. As a necessity of life, clothing often sensitively reflects the fashion, the characteristics of the times and, for young people, their lifestyle. Intelligent, cultural and sustainable clothing has become the mainstream development trend of the clothing industry. Based on the theory of reasoned action, this research takes online clothing stores as the research object and constructs the effect model between e-store image, consumer attitude and intention, and verifies the impact of e-store image on consumers' intention in the two important stages of information search and purchase. By distributing online questionnaires to consumers of online clothing stores, 823 questionnaires were finally effectively recovered. The hypothesis was verified using a multi-statistical analysis and structural equation model. The findings showed that for young consumers, in the information search stage, the three dimensions of e-store image, which are information, atmosphere and convenience, positively affect consumers' search intention. Meanwhile, in the purchase stage, the three dimensions of e-store image, which are enjoyment, uncertainty and service, have a significant impact on consumers' purchase intention. Particularly, consumer attitude plays a mediating role between different dimensions and intentions. According to the constructed consumer attitude model, we should propose these sustainable marketing suggestions for online clothing stores: First of all, online clothing store operators should improve the atmosphere, convenience and information image of electronic stores, establish a sustainable image, and enhance young consumer's identity. Secondly, using multiple marketing methods, we can establish an unimpeded one-to-one interaction with young customers through the current popular live broadcast marketing, improve the enjoyment of stores, more specifically implement marketing strategies, and strengthen the sense of sustainability.

**Keywords:** TRA; e-store image; search attitude; purchase intention; sustainable marketing

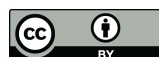

## 1. Introduction

The progress of information technology has triggered the rapid development of online shopping in China. In 2020, China's e-commerce transaction volume reached 37.21 trillion yuan. In 2021, the online retail sales reached 13.1 trillion yuan and the online retail sales of physical goods reached 10.8 trillion yuan. The online retail sales of physical goods accounted for 24.5% of the total retail sales of social consumer goods and contributed 23.6% to the growth of the total retail sales of social consumer goods. The development of online shopping has brought explosive growth in the number of e-stores. Offline retail enterprises have launched online one after another. Meanwhile, online brand stores are becoming more and more mature, and popular online stores have sprung up; they are becoming the most important stores of enterprises. In particular, the online sales of clothing, shoes and bags far exceed other categories. According to the 2016 China E-commerce Vitality Report of iResearch Consulting, 60.5% of online consumers bought such goods in 2015. In 2017, among the top five categories that Chinese consumers have purchased in the last six

months, clothing, shoes and hats topped the list, reaching 73.9%. With the economic development, the purchase frequency of clothing has gradually increased, becoming a new and fast consumer goods trend. For example, the fans' number for the Adidas Tmall flagship e-stores is close to 20 million. The transactions for the Tmall Double 11 online shopping Carnival in 2018 exceeded 1 billion yuan, which had become the largest single store in the world. E-stores provide consumers with more choices for online shopping. There are more than three million clothing stores on Taobao.com.

Within such a background, when consumers make online shopping decisions, images of e-stores and factors such as style positioning, layout and navigation, commodity preparation, pricing and promotion, display and color matching, etc., may act on consumers and affect their online shopping decisions. These images together constitute the e-store image, which is a multi-dimensional structure.

The decision-making process of consumers is a dynamic process, which consists of two important stages: information search and purchase. The multiple dimensions of e-store image have different mechanisms in the two stages of online shopping search and purchase. Information search could maximize the utility of a purchase, reduce choice uncertainty or regret, and/or satisfy curiosity about a purchaser [1]. In the search phase, the quality of information perceived by consumers is very important [2]. Visual attractive design and well-organized website structure are also important attributes. Mathwicka, et al. [3] proposed that the website atmosphere is visual attraction, including the display attraction, aesthetic requirements and overall appearance of the website, which has a positive role in promoting information search. Generally speaking, the greater the risk perceived by consumers, the more extensive the information search before purchase. Because online shopping is a new mode of shopping that contains a variety of new perceived risks, consumers may pay more attention to information search. In the purchase stage, the attributes of goods preparation, safety, service quality, convenience of purchase, reliable and timely delivery of goods are considered as important attributes, while attributes such as risk or uncertainty are resistance factors encountered by consumers in the purchase phase [4].

In addition, many scholars have used the theory of reasoned action (TRA) to study the attributes and images of e-stores. It is also found that consumers have formed different e-store attitudes and buying behaviors after perceiving different e-store images [5–9]. Their research shows that consumers can get an image by judging the attributes of online stores, thus deciding whether to buy. In other words, e-store image is an important antecedent variable affecting consumer attitudes [10], so we can therefore construct the relationship of "E-store Image-Consumer Attitude-Online Shopping Behavior". According to the theory of consumer decision making, different dimensions of e-store image have different effects in the two stages of consumer decision making, which are dynamic and simultaneous. Therefore, it is not complete and accurate to verify the influence of a certain image dimension at a certain stage. This paper introduces the two stages of search and purchase into the e-store image effect model to form two paths for the influence of different dimensions of e-store image on consumers' attitude and behavior, which is more practical.

The Internet has penetrated into all levels of life through the development of portals and search websites, social networking and mobile internet. According to the 2018 Undergraduate Consumption Insight Report of iResearch, contemporary college students have grown up in a more mature internet era and become the first generation of internet natives. At the same time, according to the 41st Statistical Report on the Development of China's Internet released by the CNNIC, the internet users in China are mainly 10–39 years old, accounting for 73% of users, acknowledging that this figure includes those with a stable career structure and the largest student group. The expenditure on clothes is the main part of young consumers' online shopping expenditure. Their online shopping behavior is the focus of this study.

With the continuous emergence of global warming and excessive waste of resources, especially in the clothing industry, sustainable clothing has become a fast-moving goods trend [11]. Intelligent, cultural and sustainable clothing has become the mainstream de-

velopment trend of the clothing industry. As a necessity of life, clothing often sensitively reflects the fashion, characteristics of the times and people's lifestyle, especially for young people. Young people usually lead fashion trends and have high requirements for clothing consumption and updating. It is more meaningful to study their e-store image of clothing and specify sustainable marketing strategies. A good image of e-stores is a beautiful picture in the minds of consumers. It not only symbolizes the strength of online retailers, but also attracts consumers to constantly browse and patronize online stores. With the development of online shopping, the competition between online stores is becoming increasingly fierce. As one of the antecedents that affect consumers' online search and purchase, how to identify and play the online store impression effect has been widely investigated by marketing scholars and online retail business operators. Therefore, it is of practical significance to establish the relationship model of e-store image, consumer attitude and consumer intention, reveal the mechanism of the impact of e-store image on consumer attitude and intention in the two stages of search and purchase, and propose corresponding sustainable marketing strategies for online retail enterprises according to the verification results. Thus, we propose the research questions:

RQ1. For online clothing stores, what is the impact mechanism of different dimensions of e-store image on consumer intention?

RQ2. In the two stages of search and purchase, which dimensions of e-store image affect consumer attitudes?

RQ3. What role does consumer attitude play in the relationship between e-store image and intention?

With these research questions, we aim to find the impact mechanism of e-store image on consumer attitudes and intentions.

The paper is organized as follows: Section 2 provides the theoretical background of this study. In Section 3, the research model and hypotheses are proposed. In Section 4, the methodological strategy and the results are presented. Section 5 provides the results of data analyses. Finally, Section 6 presents the discussion, and Section 7 provide conclusions, implications, limitations and future research, etc.

## 2. Theoretical Background

### 2.1. Theory of Reasoned Action

According to the theory of reasoned action (TRA), behavioral intention is formed by the attitudes behind the performing of a specific behavior and the subjective norms that lead to the actual behavior [12]. This subjective norm is formed on the basis of considering the availability of resources and opportunities. An attitude toward a particular behavior reflects a person's interest in performing that characteristic behavior. Generally speaking, the stronger a person's will, the more likely he is to perform such behavior. Behavioral intention is a very powerful predictor of actual behavior [12].

This model has been applied and tested by outstanding researchers in many situations, especially in the field of consumer behavior [13,14]. Based on the TRA as well, Vijayasarathy [15] pointed out that product perception, purchase experience, customer service and consumption risk are four significant beliefs that jointly determine consumers' online purchase attitude. Saleem et al. [16] identified important factors such as perceived awareness of security, perceived usefulness, personal innovativeness and perceived ease of use in purchasing, together with the effects of these factors on online purchasing intentions and the mediating role of consumer attitudes toward online purchasing, using the TRA model as well. In the research on e-stores, the operation of e-stores relies on the hardware and software quality of the website, which is a complete information system. Therefore, the early research on e-stores was conducted from the perspective of information systems. Information quality, website response time and system accessibility are the elements of information system quality. These elements affect consumer behavior beliefs, and then affect attitudes and behaviors [5]. An e-store is not only an information system, but also a retail terminal. The factors that affect consumers' use of shopping websites include not only the

quality attributes of information systems, but also the important marketing attributes of retail facilities or retail stores, such as goods and services [8]. Each element of e-store image can only explain 30% of online shopping. The unique personal factors of consumers are also another important factor to explain online shopping attitudes [6]. According to the TRA, website design, consumer characteristics (such as age, gender, social status and other personality characteristics), commodity characteristics and purchase purpose are all external variables. Only internal psychological variables such as behavior beliefs and subjective norms can indirectly affect consumer behavior. After consumers perceive different e-store images, they form different e-store attitudes and buying behaviors [5–7,9].

*2.2. E-Store Image*

E-store image, also known as an online store image or virtual store image, is based on the concept of store image. Store image is the result of consumers' perception of the functional and psychological attributes of stores [7]. Store image is described as a structure composed of multiple dimensions, which is "a structure with functional components connected together". In other words, store image is defined as the sum of perception of many store attributes. Functional attributes include commodity selection, price range, credit policy, store layout, negotiation and comparison. Instead, psychological attributes include emotional responses such as belonging, warmth, friendliness, excitement or interest.

Similarly, e-store image is the way in which a store is defined in the minds of customers, partly by its functional quality and partly by its psychological attributes, as well as by consumers' complex perception of the functional attributes of e-stores. E-store image is an important antecedent variable of consumers' online shopping attitude and behavior. On the measurement of e-store image, there are some differences in academic circles [17,18]. There are both similarities and differences in the dimensions of e-store image and store image. Both are not a single dimensional structure, but a complex structure composed of multiple dimensions. At the same time, the website is an important medium for online retail stores to spread information and conduct transactions. E-stores are virtual stores relying on website hardware and software. Therefore, some researchers thought that we could measure e-store image from the dimension of store image [19] and they used the same dimension as the physical store image when measuring e-store image [20]; other researchers measured e-store image from the perspective of website or information system quality [17], or selected several suitable online store images in the store image dimension for measurement [19]. For example, four dimensions of baby food e-store image, e-store design image, order fulfillment image, communication service image, and security image, determines purchase intentions [21]. Through the research on 40 articles about the image of online stores from 1999 to 2021, which are all influential journal papers including 26 SCI, SSCI and CSSCI research results, avoiding multiple articles divided by one scholar using the same dimension and also comprehensively considering the repeated citation of literature and new research, this study lists important dimensions and makes statistics. Finally, this paper considers that e-store image is a multidimensional construct composed of six dimensions, namely information, enjoyment, convenience, atmosphere, uncertainty and service.

Information refers to the quantity and quality of information provided by e-stores. The quantity of information reflects the quantity of information provided by channels, and the quality of information reflects the depth or professionalism of information. Information quality includes the accuracy, reliability, integrity, timeliness, timeliness, availability and understandability of information. E-store enjoyment was proposed by van der Heijden and Verhagen [22], which is an important internal motivation for consumers to use online shops. E-store enjoyment measures the amount of fun, pleasure and attraction that online shoppers perceive in online stores. Convenience means to reduce the time and effort spent by consumers in the transaction process, including easy use of online stores, store layout, search navigation and other aspects [23]. E-store atmosphere can also be called e-store aesthetics, which is the consumer perception of the atmosphere created by the e-store

design style and visual appeal [24]. Uncertainty is that consumers may bear psychological and emotional costs when shopping in a store. These psychological costs usually come from risk perception. Customers predict that something will be wrong or not as good as expected, which increases the resistance of online shopping. Service is often chosen to measure the service quality of online stores, including the overall customer service level, delivery speed and payment method. In addition, the interaction and communication between online stores and consumers is also an important service content [22,25].

### 2.3. Consumer Attitudes

Attitude is a person's overall evaluation of people (including himself), objects and problems, and it is a basic factor affecting consumer behavior. A person's attitude is described as how favorable or unfavorable it is to certain judgment objects (such as ice cream, work or a website), or how positive or negative it is in general, and reacts to the object or such objects [26]. This definition shows that attitude is not an obvious behavior, but a personality that affects behavior [27]. People first form an attitude towards a certain behavior, then develop an intention to perform the behavior, and finally perform the behavior [6]. Consumer attitudes always play a mediating role between external factors and consumers' behavior [28].

There are many types of attitudes according to different evaluation objects. Marketing scholars are more concerned about consumer attitudes, such as commodity attitude, advertising attitude, brand attitude, website attitude, etc. In this paper, according to the different decision-making stages of consumers, we pay attention to e-store attitude, and divide it into e-store search attitude and e-store purchase attitude.

Information search is the most important factor affecting online purchase. Information search attitude represents consumers' positive or negative evaluation of online search behavior. When consumers have a positive attitude towards retailers, they may show greater willingness to search for product information from retailers. Search intention plays a central role in predicting future purchase intention, so search attitude is the most valuable research tool to predict consumers' online purchase possibility [29]. At the same time, consumers' attitude towards the Internet will affect their online information search behavior [9] and online purchase behavior [30].

Online purchase attitude is defined as consumers' positive or negative feelings about online purchase behavior [31]. According to the attitude theory, the willingness to buy online is mainly explained by attitude. Online purchase attitude has a positive impact on online search intention [29,30]. Accordingly, online purchase attitude is the factor that determines online purchase intention. In order to successfully attract internet consumers, online retailers must learn more about consumers' attitudes towards online purchases, online purchase intentions and their antecedents. It is not only valuable for practitioners responsible for implementing and deploying online retail, but also attractive to researchers.

### 2.4. Consumer Intention

Intention refers to the probability of an individual performing a specific action and the possibility of an individual performing a specific action [12]. The intention indicates the motive composition of the behavior, which is the degree of consciousness of one's efforts to complete a certain behavior. According to the TRA, the intention is formed by the attitude behind the performing of a specific action and the subjective norms that lead to the actual behavior. Therefore, instilling a correct or expected behavioral belief and increasing the correlation between expected outcomes and behaviors will increase the chance of actual behaviors. Generally speaking, behavioral intention is a very powerful predictor of actual behavior. The stronger a person's intention, the more likely he is to perform such behavior [12]. In marketing, intention means that customers are willing to try, buy, adopt or oppose goods, brands, services or stores, and are willing to repeat purchases, or buy more in the future, or are willing to recommend stores to others [32]. Considering that information search and purchase are the two most critical stages in the process of consumer

purchase decision-making, consumer intention can also be divided into search intention and purchase intention [33].

Search intention refers to the possibility that consumers use online stores to search for commodity information. It is an important predictor of the actual search behavior of online stores and reflects the motivation and demand of consumers to search for information through online stores [34]. Consumers search the price and content information of the goods displayed on the seller's website. The Internet has had a significant impact on search technology. It is becoming more and more important to understand the best search strategy of consumers for information [35]. Search intention mainly depends on the power of the internet information search engine [36] and its centralized information content. Online information search refers to the behavior of users in the process of acquiring and confirming relevant knowledge or information through the Internet or mobile internet for specific situations [37].

As another form of intention, purchase intention refers to the possibility of consumers to purchase goods or services, which is used to predict purchase behavior. Previous studies have identified a positive relationship between purchase intention and purchase behavior. E-store purchase intention is an important predictor of the actual purchase behavior of e-stores, which refers to the standard evaluation results of consumers for website quality, information search and commodity estimation [38]. The purchase intention of online stores reflects the motivation and demand of consumers to purchase through e-stores.

Additionally, the relationship between search intention and purchase intention has been confirmed using the online pre-purchase intention model established by [29]. Users who are more willing to use the Internet for information search may have greater intention of purchasing goods through the Internet [39]. Klein [40] proposed in the information search economic model that consumers would choose the lowest-cost way to search for information and purchase goods or services. Ratchford et al. [41] also found that when consumers buy goods online, they may search for more information through the Internet. Online purchase intention is an increasing function of the intention to search for commodity information online [41]. Moreover, frequent internet information searching and browsing have led to frequent internet purchases [36]. Similarly, online purchasers may spend more time on the Internet than non-purchasers. This indicates that the length of time the Internet is used for information search affects online purchase behavior [42]. The positive relationship between search intention and purchase intention has also been confirmed in previous studies on clothing goods [29,30].

## 3. Research Model and Hypotheses

### 3.1. Conceptual Framework

Based on the theoretical basis of this research, this research combines e-store image (including information, atmosphere, convenience, enjoyment, uncertainty and service), consumer attitude and intention to establish the research model based on the TRA paradigm, as shown in Figure 1.

### 3.2. Research Hypotheses

3.2.1. The Influence of E-Store Image on Search Intention

Consumers' e-store image starts from opening and browsing the web. Website content plays a very important role in the process of influencing consumers' purchase decisions. Liu and Arnett [43] examined the factors that enhance the success and opportunities of commercial websites and identified four factors that affect the success of websites. The first is the quality of information and service. Richer information (more extensive and high-quality) can help consumers make better purchase decisions and achieve a higher level of online satisfaction. Although there is some overlap between information and search convenience, many authors distinguish this dimension from ease of use [44]. The perceptual information is generally based on the quantity and quality of the transmitted information, and also includes the option of being able to compare [45]. The quantity of information

reflects the quantity of information provided, and the quality of information reflects the depth (or professionalism) of information. The literature states that consumers of millennials, called "the internet generation", will always use the Internet to obtain official information [46]. Before purchasing goods, consumers search for brand, diversity, price, quality and other information. When they perceive that e-stores provide more relevant information, consumers can make decisions more easily and quickly, which can greatly increase consumers' experience and their intention to search for information.

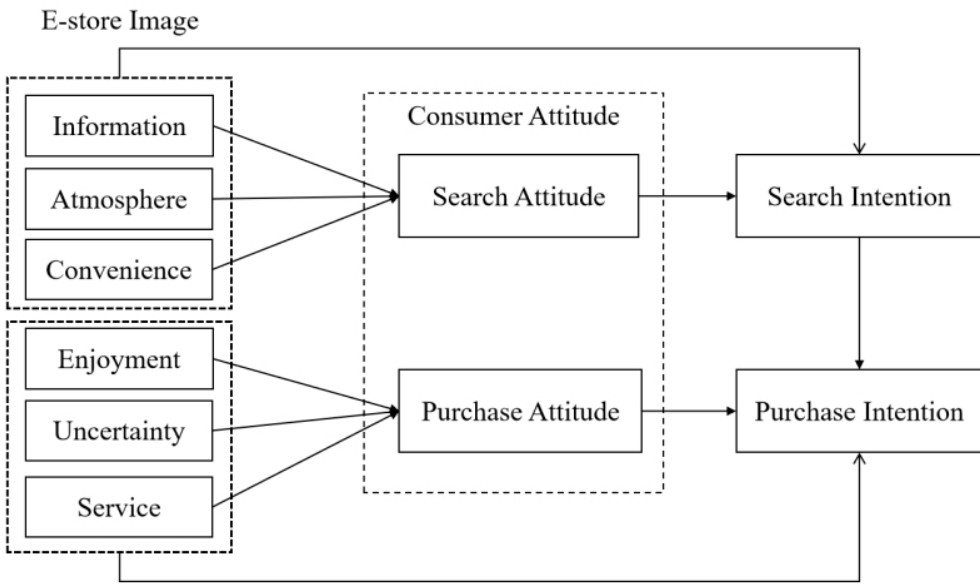

**Figure 1.** Conceptual framework.

In the search phase, visualization, attractive design and well-organized structure are important attributes of e-commerce websites. In a virtual world, the purpose of store layout is to create an environment that fascinates customers, entices them to spend more time in the store [47]. Eroglu et al. [48] defined e-store atmosphere as the sum of visual and audible clues experienced by all online shoppers, but lacking sensory attraction and excluding touch, smell and taste. Some scholars also define the website atmosphere as visual attraction, including the display attraction, aesthetic requirements and overall appearance of the website [3]. In this study, e-store atmosphere cues include two dimensions: visual composition and response; website design and visual attraction. The atmosphere of online retail websites or stores is used as a stimulus to stimulate consumers' pleasant and awakened emotions, and then affect consumers' attitudes and make them respond accordingly [48]. Therefore, it can be predicted that a good e-store atmosphere can improve consumers' intention to search e-stores.

In addition, another factor affecting the e-store image is the perception of convenience. Millennials are a generation that expects processes to be effortless [46]. Online shopping meets the convenience needs of young consumers to shop anytime and anywhere. More convenient browsing media, fast searches and convenient navigation help consumers reduce the cost of obtaining commodity information before purchase. Janiszewski [49] classified search behavior into two types: goal oriented and exploration oriented. Target-oriented consumers have solid shopping plans in their minds, so clear product classification, store navigation and search can enable them to quickly locate the goods they want. Exploration-oriented consumers have no fixed plan in mind. The purpose of their search is only to browse the store. Convenience will add comfort to their browsing. Therefore, convenience has greatly improved consumers' intention to search. Therefore, this study proposes the following assumptions:

**H1a.** *Information has a significant positive impact on consumers' e-store search intention.*

**H1b.** *Atmosphere has a significant positive impact on consumers' e-store search intention.*

**H1c.** *Convenience has a significant positive impact on consumers' e-store search intention.*

3.2.2. The Influence of E-Store Image on Purchase Intention

Enjoyment is considered to be an important internal motivation for e-store use. It was proposed by [22] in their e-store image research, and it is considered to be an important driver of perceived ease of use [23]. Some scholars believe that the practical quality (ease of use and convenience) of online shopping is an important predictor of attitude and purchase intention, but the enjoyment of the Internet has at least the same status in determining these aspects [3]. van der Heijden and Verhagen [22] were the first to study the customers of two Dutch online bookstores and developed an e-store image scale composed of 27 attributes, including seven dimensions: e-store usefulness, enjoyment, ease of use, store style, familiarity, trust and settlement performance. Among them, e-store enjoyment has a significant impact on online purchase intention. Verhagen and van Dolen [50] proposed that enjoyment has a significant impact on impulse buying behavior. Especially, millennials are looking for hedonic value in online stores [46].

Consumers' buying behavior can be regarded as risk taking. Any choice situation contains two risks: the uncertainty of output and the uncertainty of result [51]. Cox and Rich [52] defined perceived risk as the total uncertainty perceived by consumers in a specific purchase situation. Risks include financial, commodity performance, physical, social and psychological risks, as well as loss of time or convenience [53]. When consumers buy goods online, they may be afraid of losing money, and the goods they buy may not be as expected or may not meet their needs, becoming a waste of time and causing psychological discomfort. Millennials are also known as digital natives. They demonstrate higher levels of trust, are more tolerant, support social causes, and are socially responsible companies [46]. Many studies have shown that uncertainty negatively affects purchase intention [54]. High perceived risk prevents consumers from shopping online [55].

In addition to the above factors, the service level accepted by customers is usually regarded as a component of store image or attitude [32], which is a decisive factor in creating customer satisfaction [56]. Online services mainly have four components: efficiency, reliability, performance and privacy, which constitute the core dimensions of services [57]. Online shoppers focus on the more practical ease of use and speed. They only need help when there are problems, or when they have questions that need answering. In those cases, what they usually need is a quick response. Online service quality plays a very important role in the decision-making process of consumers' purchase. Fast and safe delivery, return and exchange services increase customers' online purchase [8]. Therefore, this study proposes the following assumptions:

**H2a.** *Enjoyment has a significant positive impact on consumers' e-store purchase intention.*

**H2b.** *Uncertainty has a significant negative impact on consumers' e-store purchase intention.*

**H2c.** *Service has a significant positive impact on consumers' e-store purchase intention.*

3.2.3. Mediating Effect of Consumers' Attitude

Attitude is a person's overall evaluation of people (including himself), objects and problems. It is a basic factor affecting consumer behavior. According to the theory of reasoned action (TRA), e-store images affect consumers' attitudes and thus determine purchasing behavior. Thus, attitude plays an intermediary role between image and intention [16]. This intermediary role has been confirmed in the related fields of and research. Brand image helps consumers to retrieve and process information. It provides reasons for purchasing and using brand goods and produces attitudes related to goods [58]. Therefore, the better the brand image, the more positive the attitude towards brand goods, thus enhancing consumers' purchase intention [59]. Considering the online retail environment, research has shown that some consumers prefer to buy goods with established brands on-

line [60]. Consumers not only give more favorable ratings and more positive attitudes to stores with established brands, but also have greater intention to buy online.

Consumers' attitude towards the Internet is an important determinant of commodity information search using the Internet. Helander and Khalid [61] found that a positive attitude towards e-commerce has a significant impact on internet shopping. Klein [40] assumed that the Internet may affect information search behavior because it is more convenient and accessible. A positive attitude towards the Internet will increase the behavior of collecting information on the Internet. There is a positive relationship between attitude and information search behavior. When consumers have a positive attitude towards retailers, they may be more willing to search for product information from retailers. The empirical results show that internet attitude has a positive impact on internet search intention [29,30]. The more positive the attitude of e-stores, the greater intention to search for product information through e-stores.

The same is true at the purchase stage. Some past studies have shown that attitude towards online shopping is positively correlated with online shopping intention [29]. The positive attitude towards internet shopping significantly enhanced the intention to use the Internet to buy. Attitudes towards internet shopping have a positive impact on the purchase intention through the Internet. Moreover, consumers who have a better attitude towards online shopping are more willing to buy clothes online. Watchravesringkan and Shim [30] also confirmed the positive causal relationship between online shopping attitude and online shopping intention for clothing products. Similarly, Yoh et al. [62] found that internet clothing shopping attitude affects internet clothing purchase intention.

The attribute-attitude-behavior method has been applied to previous studies on the impact of retailer characteristics on shoppers' attitude and behavior. According to this method, Berthon et al. [63] concluded that consumers have a more positive and friendly attitude towards websites with higher interactivity than websites without interactivity. Elliott and Speck [64] used an inductive method to study whether consumers' attitudes towards a retail website are strongly related to image attributes. Scholars measured the six components of e-store image and found that five of them, namely ease of use, product information, entertainment, trust and timeliness, can indeed explain most of consumers' attitudes. van der Heijden, Verhagen and Creemers [7] proposed that the four attributes of e-stores (usefulness, enjoyment, trust and settlement performance) significantly affected consumers' purchase attitude and intention to buy books online. It is particularly important for retailers to create an image that arouses consumers' interest, affects their attitude towards e-stores and commodities, and ensures or modifies their behavior.

Mediating Effect of Search Attitude

Consumers' e-store search attitude refers to consumers' positive or negative evaluation of online search behavior. For example, the e-store search is very attractive, and consumers like to browse the e-store irregularly, which is the attitude brought about by the positive e-store search image. Liu and Arnett [43] found that a well-designed e-store can effect more customer revisits and improve attitudes towards e-stores and their products. The content and authenticity of the picture display (whether there are models or not) are important predictors of customers' attitudes towards e-stores, and their attitudes are also strong predictors of behavioral intentions. E-stores can quickly and accurately get the information they want to search. They are very convenient, have aesthetic enjoyment, and can master the search experience for fashion trends, which improves consumers' search attitude towards e-stores, thus affecting the search intention. Based on the above inference, this study proposes the following assumptions:

**H3a.** *Search attitude plays an intermediary role between information and search intention.*

**H3b.** *Search attitude plays an intermediary role between atmosphere and search intention.*

**H3c.** *Search attitude plays an intermediary role between convenience and search intention.*

Mediating Effect of E-Store Purchase Attitude

Online purchase attitude is defined as consumers' positive or negative feelings about online purchase behavior [31]. For example, if it is a good choice to buy in a certain e-store, consumers are willing to repeat their purchases in this e-store, and they also think it is attractive to buy in this e-store. These are attitudes brought about by the positive image of e-store purchase. So, if e-stores can enhance the fun of the store, provide services to meet consumers, and reduce the uncertainty in all online shopping, they enhance consumers' purchase attitude and purchase intention. Based on the above inference, this study proposes the following assumptions:

**H4a.** *Purchase attitude plays an intermediary role between enjoyment and purchase intention.*

**H4b.** *Purchase attitude plays an intermediary role between uncertainty and purchase intention.*

**H4c.** *Purchase attitude plays an intermediary role between service and purchase intention.*

## 4. Materials and Methods

### 4.1. Questionnaire Design

Based on the research structure, a questionnaire was used to survey consumers with previous purchase experience in online clothing stores. This study involves five core constructs: e-store image, search intention, purchase intention, search attitude and purchase attitude. The questionnaire items were modified from previous scales. Measurement items for six dimensions of e-store image were adapted from Jiang and Wu [65]. Measurement items for the search attitude and purchase attitude were adapted from Yoo and Donthu [66] and van der Heijden and Verhagen [6]. Measurement items for search intention were adapted from To et al. [67]. Purchase intention was measured using three items adapted from Yun and Good [25]. Three doctoral students translated the items and then studied and discussed with experts, repeatedly modifying the semantics to adapt it to the research situation. Finally, all items were measured on a 5-point Likert scale, ranging from 1 (not agree at all) to 5 (absolutely agree). The variable and the measurement scale are shown in Table 1.

### 4.2. Sampling Procedure

Based on the research purpose and motivation, the research structure and questionnaire were developed. Before the formal questionnaire was issued, twenty-six consumers with different educational levels and different professional backgrounds were invited to fill in the questionnaire, so that they could fill in the questionnaire in a relaxed state as much as possible, and the filling time of each questionnaire was recorded one by one. According to the test, the time for filling in this questionnaire is between 90 s and 150 s. This study also conducted pre-test analyses; the Cronbach's alpha coefficient of the overall scale was greater than 0.7, indicating that the internal consistency and reliability of the scale were good.

The formal survey adopts the principle of convenient sampling, with the answers provided online through WeChat and via forwarding the questionnaire link and scanning the QR code. The respondents can complete the questionnaire by filling in the screening items set in the questionnaire (whether there are online clothing stores they collect or care about) and writing the name of the store. The formal questionnaire will be issued from 30 November 2021 to 20 December 2021. A total of 1335 questionnaires were distributed to young people aged 19 to 24. Through the 'Are there any e-stores that are collected or frequently followed?' screening item, 971 young people had online clothing stores that they collected or paid close attention to, accounting for 72.73%, while the remaining 364 young people did not collect or pay attention to online clothing stores, accounting for 27.34% of the total questionnaires. Among the 971 people with "yes", 148 invalid questionnaires were excluded, and 823 valid questionnaires were finally recovered. The effective rate of questionnaire recovery was 84.76%.

**Table 1.** The variable and the measurement scale.

| Variable | Measurement Scale |
|---|---|
| Service | S1. The e-store has low freight |
| | S2. The e-store delivers goods quickly |
| | S3. The customer service attitude of the e-store was good |
| Information | I1. The information provided by the e-store is complete |
| | I2. The e-store provides sufficient product information |
| | I3. The e-store provides reliable information |
| Enjoyment | E1. Shopping in this e-store is exciting |
| | E2. Shopping in this e-store is pleasant |
| | E3. Buying in this e-store is a process of enjoyment |
| Convenience | C1. It doesn't take much brain to Shop in this e-store |
| | C2. The e-store's search ability is fast and accurate |
| | C3. The navigation of the e-store is clear |
| Atmosphere | A1. The e-store has a unique design style |
| | A2. The e-store is pleasant to the eye |
| | A3. The pictures of the e-store are attractive |
| Uncertainty | U1. You may buy poor quality goods in this e-store |
| | U2. You may buy counterfeit goods in this e-store |
| | U3. The risk of buying in this e-store is very high |
| Search attitude | SA1. It's a good idea to search for product information in this e-store |
| | SA2. Searching for product information in this e-store is very attractive to me |
| | SA3. I have a positive attitude towards searching for product information in this e-store |
| Search intention | SI1. I will search for the clothes I want to buy in this e-store in the future |
| | SI2. If I need to buy clothes, I will first consider searching for product information in this e-store |
| | SI3. In the future, I may search for product information in this e-store |
| Purchase attitude | PA1. I have a positive attitude towards buying clothes in this e-store |
| | PA2. Buying clothes in this e-store is very attractive to me |
| | PA3. I think it's a good idea to buy clothes in this e-store |
| Purchase intention | PI1. I will buy clothes in this e-store in the future |
| | PI2. If I need to buy clothes, I will first consider buying in this e-store |
| | PI3. I will recommend this e-store to my relatives and friends |

(Note: "E-store image" is the overarching variable label spanning Service, Information, Enjoyment, Convenience, Atmosphere, and Uncertainty.)

After screening items, the respondents need to select an online clothing store for collection or continuous attention for evaluation. Before evaluation, they need to give the name of the e-store. There is a strong correlation between store name and store image, because store name is an important clue in forming store image [68]. The respondents wrote down hundreds of e-stores' names. Among them, there are online flagship stores of well-known clothing brands, such as UNIQLO, Zara and Metersbonwe. There are also well-known online brand clothing stores in China, such as Handu Group and Miss CocoLi. At the same time, there are also some self-owned brand clothing stores founded by internet celebrities, such as the Goblin's Pocket and My Happy Wardrobe, and many private customized clothing stores, such as Wine, Red and Pickle Customized Women's Clothing and Small Tomato Customized Clothing, as well as characteristic niche fashion stores, such as

Meizi's Familiar Literary and Artistic Retro Fashion Women's Clothing, Hong Kong Style Chaoren Museum, Heygirl Black Brother, and so on. Online clothing stores identified by consumers include both B2C Tmall flagship stores and C2C Taobao stores. The investigation scope is relatively wide, which is in line with expectations.

### 4.3. Ethical Considerations

Consumers voluntarily participated in the study and provided their consent by clicking on a button placed at the beginning of the online survey. During the opening words of the questionnaire of this study, all participants have been fully informed that anonymity is assured, and that the data would only be used for scientific research. This study ensured the anonymity, privacy and security of the respondents.

### 4.4. Data Analyses

In this study, SPSS 20.0 and AMOS 24.0 software were used for multivariate statistical analysis, including exploratory factor analysis, confirmatory factor analysis, regression analysis, bootstrap and other methods to test the impact of online store impression on willingness and the intermediary role of consumer attitudes.

## 5. Results

### 5.1. Demographics

A total of 823 valid samples were composed of 20.5% men and 79.5% women. The proportion of women is significantly higher than that of men. On the one hand, more male samples were eliminated from the screening item 'Are there any e-stores that are collected or frequently followed?', indicating that men like collecting from e-stores less than women. On the other hand, women consumers are more impressed with the store than men consumers when buying clothing. All samples are between 19 and 24 years old. From the perspective of the education level, 94% of them are college students, which is the main group of this study. Among them, 37.7% said they would browse online stores whenever they were free, while 10% and 15.3% said they would browse three to five times a week and one to two times a week. A total of 28.6% of people only browse when they need to buy. In terms of online shopping frequency, 32.2% of young people buy online once a month, 28.4% buy online half a month and 27% buy online more than once a week.

### 5.2. Reliability and Validity of the Measurement Instrument

SPSS 20.0 software was used to analyze the reliability of the overall sample data. As shown in Table 2, the Cronbach's alpha coefficient of the overall scale was greater than 0.7, and the scale had good reliability and high internal consistency.

The KMO and Bartlett's test results for the effective samples (N = 823) show that the KMO value of the scale is 0.870 > 0.7, which has passed Bartlett's spherical test, and the chi square value is 6450.926, D$f$ = 153, sig = 0.000, which indicates that the sample data is suitable for exploratory factor analysis.

The validity test of the scale mainly includes aggregation validity and discrimination validity. The aggregate validity of the scale was evaluated using the standardized factor load, composite reliability (CR) and average variance extracted (AVE). All the standardized factor loads of other items were greater than 0.5 and significant at 0.001. The highest composite reliability is 0.877, the lowest is 0.772, both greater than 0.6; the AVE values for all constructs are greater than 0.5, the highest is 0.704. Therefore, aggregation validity is supported. By comparing the AVE and latent variable correlation coefficient matrix to judge the discriminant validity, the whole model has good discriminant validity [46].

The discriminant validity was determined by comparing the correlation coefficient matrix of the AVE and latent variables. It can be seen from Table 3 that the arithmetic square root of the AVE value on the diagonal is significantly greater than its correlation coefficient with other variables. Therefore, the overall model of this study has good discriminant validity.

**Table 2.** Results of reliability analysis.

| Dimensions/Variables | Indicator | Standard Loading [a] | Cronbach's α | AVE | CR |
|---|---|---|---|---|---|
| Atmosphere | A1<br>A2<br>A3 | 0.732<br>0.840<br>0.704 | 0.816 | 0.579 | 0.804 |
| Enjoyment | E1<br>E2<br>E3 | 0.777<br>0.805<br>0.794 | 0.837 | 0.627 | 0.835 |
| Uncertainty | U1<br>U2<br>U3 | 0.829<br>0.874<br>0.813 | 0.825 | 0.704 | 0.877 |
| Information | I1<br>I2<br>I3 | 0.751<br>0.817<br>0.781 | 0.844 | 0.614 | 0.827 |
| Convenience | C1<br>C2<br>C3 | 0.746<br>0.820<br>0.694 | 0.741 | 0.570 | 0.798 |
| Service | S1<br>S2<br>S3 | 0.777<br>0.830<br>0.612 | 0.645 | 0.556 | 0.787 |
| Search attitude | SA1<br>SA2<br>SA3 | 0.798<br>0.829<br>0.834 | 0.874 | 0.673 | 0.861 |
| Search intention | SI1<br>SI2<br>SI3 | 0.784<br>0.651<br>0.745 | 0.808 | 0.531 | 0.772 |
| Purchase attitude | PA1<br>PA2<br>PA3 | 0.828<br>0.806<br>0.846 | 0.877 | 0.684 | 0.866 |
| Purchase intention | PI1<br>PI2<br>PI3 | 0.884<br>0.714<br>0.619 | 0.794 | 0.558 | 0.788 |

[a] All standard loadings were significant at $p < 0.001$.

**Table 3.** Results of discriminant validity testing.

| | M | S.D. | 1 | 2 | 3 | 4 | 5 | 6 | 7 | 8 | 9 | 10 |
|---|---|---|---|---|---|---|---|---|---|---|---|---|
| Information | 3.877 | 0.653 | 0.784 | | | | | | | | | |
| Enjoyment | 3.870 | 0.638 | 0.566 ** | 0.792 | | | | | | | | |
| Convenience | 3.692 | 0.656 | 0.486 ** | 0.438 ** | 0.755 | | | | | | | |
| Atmosphere | 4.007 | 0.606 | 0.467 ** | 0.486 ** | 0.468 ** | 0.761 | | | | | | |
| Uncertainty | 2.770 | 0.829 | −0.366 ** | −0.301 ** | −0.222 ** | −0.245 ** | 0.839 | | | | | |
| Service | 3.752 | 0.650 | 0.248 ** | 0.235 ** | 0.326 ** | 0.279 ** | −0.048 | 0.746 | | | | |
| Search attitude | 3.906 | 0.583 | 0.498 ** | 0.432 ** | 0.459 ** | 0.527 ** | −0.172 ** | 0.308 ** | 0.820 | | | |
| Search intention | 3.939 | 0.569 | 0.479 ** | 0.448 ** | 0.473 ** | 0.492 ** | −0.250 ** | 0.294 ** | 0.681 ** | 0.729 | | |
| Purchase attitude | 3.949 | 0.581 | 0.490 ** | 0.526 ** | 0.463 ** | 0.544 ** | −0.255 ** | 0.316 ** | 0.666 ** | 0.775 ** | 0.827 | |
| Purchase intention | 3.860 | 0.598 | 0.433 ** | 0.437 ** | 0.457 ** | 0.447 ** | −0.268 ** | 0.294 ** | 0.581 ** | 0.718 ** | 0.766 ** | 0.747 |

The value on the diagonal is the square root of AVE; all others are correlation coefficients. ** $p < 0.01$.

### 5.3. Model Fit

We tested the measures of the model fit using the AMOS in Table 4. The model was tested to fit well with χ2/df = 2.713 (standardized to less than 5), the root mean squared error of approximation (RMSEA) = 0.046 (standardized to less than 0.08), root mean square residual (RMR) = 0.032 (less than 0.05, which we consider a good model fit), incremental fit index (IFI) = 0.954, adjusted goodness of fit index (AGFI) = 0.907, Tucker–Lewis index (TLI) = 0.946, normative fit index (NFI) = 0.929, comparative fit index (CFI) = 0.954, goodness-of-fit index (GFI) = 0.925 (criterion is greater than 0.90), parsimony fit index (PGFI) = 0.748 and adjusted normative fit index (PNFI) = 0.803 (criterion is greater than 0.50) [69]. It demonstrated a good fit between the model and the data.

**Table 4.** Measures of the model fit.

| Fit Index | CMIN/DF | RMSEA | RMR | GFI | CFI | NFI | IFI |
|---|---|---|---|---|---|---|---|
| Model value | 2.713 | 0.046 | 0.032 | 0.925 | 0.954 | 0.929 | 0.954 |

### 5.4. Hypotheses Testing

After examining the validity and reliability, the SPSS was used to test the proposed hypotheses. The results of the main effect test between the three dimensions of online store image (information, atmosphere and convenience) and consumers' search intention are shown in Table 4. Information, atmosphere and convenience all have a significant positive impact on search intention. It is assumed that H1a, H1b and H1c pass the verification ($p < 0.001$). The results of the main effect test between the three dimensions of online store image (enjoyment, uncertainty and service) and consumers' purchase intention are also shown in Table 5. Enjoyment, uncertainty and service all have a significant impact on purchase intention. Above all, uncertainty has a significant negative impact on purchase intention. It is assumed that H2a, H2b and H2c pass the verification as well ($p < 0.001$).

**Table 5.** Results of hypotheses testing.

| | Standardized Coefficient | t | Sig. | VIF | $R^2$ | Adjusted $R^2$ | F | Hypothesis | Support |
|---|---|---|---|---|---|---|---|---|---|
| Information | 0.210 | 7.146 | 0.000 | 1.448 | | | | H1a | Yes |
| Atmosphere | 0.256 | 8.183 | 0.000 | 1.416 | 0.358 | 0.355 | 155.067 *** | H1b | Yes |
| Convenience | 0.198 | 6.778 | 0.000 | 1.450 | | | | H1c | Yes |
| | | Search intention is dependent variable; *** $p < 0.001$. | | | | | | | |
| Enjoyment | 0.320 | 10.486 | 0.000 | 1.162 | | | | H2a | Yes |
| Uncertainty | −0.112 | −4.907 | 0.000 | 1.101 | 0.252 | 0.249 | 91.748 *** | H2b | Yes |
| Service | 0.189 | 6.614 | 0.000 | 1.059 | | | | H2c | Yes |
| | | Purchase intention is dependent variable; *** $p < 0.001$. | | | | | | | |

### 5.5. Mediating Effect Testing

On the basis of the above two main effects, in order to test the mediating effect of consumer attitudes, this study constructed two models. The bootstrap method of the AMOS 24.0 was used for parameter estimation (sampling times N = 5000). Table 6 shows the mediating effect.

**Table 6.** Results of mediating effect testing.

| Effect Type | Effect | p | Lower 95% CI | Upper 95% CI |
|---|---|---|---|---|
| Search intention as dependent variable | | | | |
| Information → Search intention (direct) | 0.075 | 0.083 | −0.011 | 0.161 |
| Atmosphere → Search intention(direct) | 0.121 | 0.008 | 0.031 | 0.204 |
| Convenience → Search intention (direct) | 0.111 | 0.013 | 0.023 | 0.198 |
| Information → Search attitude → Search intention (indirect) | 0.186 | 0.000 | 0.122 | 0.257 |
| Atmosphere → Search attitude → Search intention (indirect) | 0.239 | 0.000 | 0.175 | 0.309 |
| Convenience → Search attitude → Search intention (indirect) | 0.134 | 0.000 | 0.066 | 0.207 |
| Information → Search intention (total) | 0.261 | 0.001 | 0.164 | 0.355 |
| Atmosphere → Search intention (total) | 0.361 | 0.000 | 0.270 | 0.450 |
| Convenience → Search intention (total) | 0.245 | 0.000 | 0.141 | 0.347 |
| $\chi 2 = 2.429$, CFI = 0.982, TLI = 0.976, NFI = 0.970, and RMSEA = 0.042. | | | | |
| Purchase intention as dependent variable | | | | |
| Enjoyment → Purchase intention (direct) | −0.055 | 0.191 | −0.133 | 0.025 |
| Uncertainty → Purchase intention (direct) | −0.066 | 0.017 | −0.125 | −0.013 |
| Service → Purchase intention (direct) | 0.007 | 0.808 | −0.056 | 0.071 |
| Enjoyment → Purchase attitude → Purchase intention (indirect) | 0.437 | 0.001 | 0.327 | 0.543 |
| Uncertainty → Purchase attitude → Purchase intention (indirect) | −0.077 | 0.048 | −0.149 | −0.001 |
| Service → Purchase attitude → Purchase intention (indirect) | 0.273 | 0.000 | 0.178 | 0.379 |
| enjoyment → Purchase intention (total) | 0.382 | 0.001 | 0.276 | 0.480 |
| Uncertainty → Purchase intention (total) | −0.143 | 0.002 | −0.221 | −0.059 |
| Service → Purchase intention (total) | 0.280 | 0.000 | 0.176 | 0.388 |
| $\chi 2 = 1.813$, CFI = 0.989, TLI = 0.985, NFI = 0.976 and RMSEA = 0.031. | | | | |

The indirect path, information → search attitude → search intention, is significant with an effect value = 0.186, $p$ = 0.000 < 0.01, and a 95% bootstrap confidence interval excluding zero. This shows that search attitude plays a significant mediating role between information and search intention. Meanwhile, the direct path, information → search intention, is not significant. This shows that search attitude plays a full mediating role between information and search intention. Hypothesis H3a was verified.

The indirect path, atmosphere → search attitude → search intention, is significant with an effect value = 0.239, $p$ = 0.000 < 0.001, and a 95% bootstrap confidence interval excluding zero. This shows that search attitude plays a significant mediating role between atmosphere and search intention. Meanwhile, the direct path, atmosphere → search intention, is also significant. The effect value = 0.121, $p$ = 0.008 < 0.05, with a 95% bootstrap confidence interval excluding zero. This shows that search attitude plays a partial mediating role between atmosphere and search intention. Hypothesis H3b was verified.

The indirect path, convenience → search attitude → search intention, is significant with an effect value = 0.134, $p$ = 0.000 < 0.001, and a 95% bootstrap confidence interval excluding zero. This shows that search attitude plays a significant mediating role between convenience and search intention. Meanwhile, the direct path, convenience → search intention, is also significant. The effect value = 0.111, $p$ = 0.013 < 0.05, with a 95% bootstrap confidence interval excluding zero. This shows that search attitude plays a partial mediating role between convenience and search intention. Hypothesis H3c was verified.

The indirect path, enjoyment → purchase attitude → purchase intention, is significant with an effect value = 0.437, $p$ = 0.001 < 0.01, and a 95% bootstrap confidence interval excluding zero. This shows that purchase attitude plays a significant mediating role between enjoyment and purchase intention. Meanwhile, the direct path, enjoyment → purchase intention, is not significant. This shows that purchase attitude plays a full mediating role between enjoyment and purchase intention. Hypothesis H4a was verified.

The indirect path, uncertainty → purchase attitude → purchase intention, is significant with an effect value = −0.077, $p$ = 0.048 < 0.05, and a 95% bootstrap confidence interval excluding zero. This shows that purchase attitude plays a significant mediating role between uncertainty and purchase intention. Meanwhile, the direct path, uncertainty → purchase intention, is also significant. The effect value = −0.066, $p$ = 0.017 < 0.05, with a 95% bootstrap confidence interval excluding zero. This shows that purchase attitude plays a partial mediating role between uncertainty and purchase intention. Hypothesis H4b was verified.

The indirect path, service → purchase attitude → purchase intention, is significant with an effect value = 0.273, $p$ = 0.000 < 0.001, and a 95% bootstrap confidence interval excluding zero. This shows that purchase attitude plays a significant mediating role between service and purchase intention. Meanwhile, the direct path, service → purchase intention, is not significant. This shows that purchase attitude plays a full mediating role between service and purchase intention. Hypothesis H4c was verified.

The types of mediation roles are summarized in Table 7.

**Table 7.** The types of mediation roles.

| Dependent Variables | Mediator Variables | Independent Variables | Types of Mediating Effect | Hypothesis | Support |
| --- | --- | --- | --- | --- | --- |
| Information | Search attitude | Search intention | Full mediation | H3a | Yes |
| Atmosphere | Search attitude | Search intention | Partial mediation | H3b | Yes |
| Convenience | Search attitude | Search intention | Partial mediation | H3c | Yes |
| Enjoyment | Purchase attitude | Purchase intention | Full mediation | H4a | Yes |
| Uncertainty | Purchase attitude | Purchase intention | Partial mediation | H4b | Yes |
| Service | Purchase attitude | Purchase intention | Full mediation | H4c | Yes |

## 6. Discussion

The results show that all of the 12 hypotheses in this study are supported. First, in the information search stage, the three dimensions of e-store image, which are information, atmosphere and convenience, positively affect consumers' search intention when these three dimensions act on consumers at the same time. The importance of each dimension from strong to weak is atmosphere, information and convenience.

Second, in the purchase stage, the three dimensions of e-store image, which are enjoyment, uncertainty and service, have a significant impact on consumers' purchase intention when they act on consumers at the same time. The importance of each dimension from strong to weak is enjoyment, service and uncertainty.

Thirdly, consumers' search attitude plays an intermediary role between online store image and search intention, meanwhile purchase attitude plays an intermediary role between online store image and purchase intention. Based on the TRA, this study divides consumers' e-store attitude into search attitude and purchase attitude. The results of the data analysis show that: Firstly, search attitude plays a partial mediating role between information, atmosphere and search intention, and plays a full mediating role between convenience and search intention. Secondly, purchase attitude plays a full mediating role between pleasure, service and purchase intention, and plays a partial mediating role between uncertainty and purchase intention. This conclusion finds that e-store image can affect search intention and purchase intention through search attitude and purchase attitude, respectively.

## 7. Conclusions, Implications, Limitations and Future Research

This study successfully verified the impact effect of all dimensions of e-store image on consumers at the same time. Existing research usually focused on the impact of individual dimensions of e-store image on consumers' purchase intention and could not recreate the real online shopping situation consumers experience. This study comprehensively verified that when all dimensions work at the same time, e-store image has a significant impact on intention, and that consumers' attitude plays an intermediary role between image and intention. The conclusion is more scientific and accurate than other studies.

This study also changes the previous separate study of consumer information search and purchase, introducing the two-stage decision-making process of search and purchase into the same effect model of e-store image. The most important stages in the decision-making process of consumers are search and purchase. Although many scholars studied consumer behavior from the two stages of search and purchase, most of them focused on the two stages of consumer behavior from the perspective of multi-channels, and compared online and offline as a whole, ignoring the differences between retail terminals in the same channel and that the final choice of consumers is not the channel but the retail terminal. This study fully considered the different roles of the six dimensions of e-store image in different stages of consumer decision-making, and better restored the real decision-making environment. It is clear that the dimensions of information, atmosphere and convenience of e-store image play a role in the search stage and have a significant impact on the search intention, while the dimensions of enjoyment, uncertainty and service of e-store image play a role in the purchase stage and have a significant impact on the purchase intention. This further clarified the varying importance of different dimensions of e-store image in different stages and promoted the research progress of e-store image.

E-store operators can formulate sustainable marketing strategies according to the conclusions of this study. The practical implications of this study mainly have the following two aspects: First, e-store operators should pay attention to the attitude at the stage of consumer information search and improve the corresponding e-store attributes in a targeted manner, so as to improve the search frequency of stores, and then increase sales. Search intention has a positive effect on purchase intention in online clothing stores [29,30]. For example, atmosphere is the most important image dimension for young consumers in the search stage. The overall decoration atmosphere of the online clothing store can directly

convey the style and characteristics of the store's clothing. E-store operators should set up professional operation and art teams of stores, pay more attention to the design of the store atmosphere environment, which includes forming a consistent overall style, color matching and aesthetic enjoyment, and updating the store layout with festivals or promotional activities to create an atmosphere, so that young consumers can enjoy the beauty of searching or browsing in the store, thus promoting search behavior and directing consumers to purchase. Second, e-store operators should grasp the key factors of young consumers when making purchase decisions to improve store performance. The enjoyment of e-stores is very important to the online purchase of the young generation of the current major consumer groups. According to the verification results, improving enjoyment of the e-store can better improve the purchase intention of young consumers. Online clothing stores can provide young consumers with good live broadcast scenes, live broadcast effects and clothing product experience. Under the live broadcast with good picture quality, young customers can quickly integrate into the shopping scene, so that consumers can feel the satisfaction and happiness of online, live broadcast shopping. This context innovates and enriches the content of interaction, provides young consumers with a novel, interesting and immersive experience, and enables consumers to obtain good emotional experience, content experience and interactive experience during the live broadcast process.

This study, as with any other study, has several limitations. First, the test of the theoretical model in this study is based on the Chinese background. At the same time, the online clothing store is selected for the study. Second, the focus of this study is on the impact of e-store images on consumers' e-store choices. The premise is that consumers have perception of e-stores. Therefore, at the beginning of the questionnaire survey, the necessary respondents should be identified. In the survey, 27.34% of the consumers have no perception of e-stores, especially male consumers. Although they have online shopping experience, they do not necessarily pay attention to the e-stores. Each online shopping behavior is a target-oriented search product, which does not form a store image, so it is not the subject of this study. This study did not explore the reasons why these consumers did not form a e-store image, nor how to make them form a e-store image and become loyal customers of the store. However, these are all very meaningful topics.

Future research can verify the model conclusion under different cultural backgrounds, different commodity categories, service enterprises and other backgrounds. Whether the universality of the research conclusion can be fully applicable to other types of e-stores or e-stores providing services needs to be further tested. Second, future studies can make a comparative analysis of "imaged" and "non-imaged" consumers, and further improve the formation mechanism of consumers' e-store image. This study did not distinguish between computer online stores and mobile online stores and used a questionnaire survey. Future research can distinguish online store terminals and use methods such as an eye movement experiment to study the impact of store atmosphere, layout, color and other impressions on consumer behavior based on visual perception.

**Author Contributions:** Conceptualization, Q.L. and X.W.; methodology, L.J.; software, L.J.; validation, Q.L. and X.W.; formal analysis, L.J.; investigation, L.J.; resources, Q.L.; data curation, L.J.; writing—original draft preparation, L.J.; writing—review and editing, Q.L.; visualization, X.W.; supervision, X.W.; project administration, Q.L.; funding acquisition, Q.L. and L.J. All authors have read and agreed to the published version of the manuscript.

**Funding:** This research was funded by the Jilin Social Science Research Planning Project, grant number: JJKH20211349SK; and Social Science Foundation Project of Jilin Province, grant number: 2022B89.

**Institutional Review Board Statement:** Not applicable.

**Informed Consent Statement:** Not applicable.

**Data Availability Statement:** The related data sets are available from the corresponding author upon request.

**Acknowledgments:** The authors sincerely thank anonymous reviewers' constructive comments.

**Conflicts of Interest:** The authors declare no conflict of interest.

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
