# Peer review of "The Impact of Clothing E-Store Image on Intention Based on Search and Purchase Phases: From the Perspective of Sustainable Marketing"

_sustainability, doi:10.3390/su15010871_

Round 1

Reviewer 1 Report

In this paper the authors discuss about online clothing stores and constructs the effect model between e-store image, consumer attitude and intention, and verifies the impact of e-store image on consumers' intention in the two important stages of information search and purchase. Using a sample of 1058 respondents, the findings showed that in the information search stage, the three dimensions of e-store image, which are information, atmosphere and convenience, positively affect consumers' search intention.

Although the paper falls into the topics of the journal, I think that the manuscript should be improved in order to meet the high expectation of the journal:

1.     Section 1. Introduction- the authors should introduce the research questions and the structure of the paper here. More, the authors should introduce more valuable references, especially from MDPI journals. For example,

2.     Section 4. Research methodology- this section is very briefly presented by the authors and also it concerns me the most. Please present the methodology of research (sampling design, data collection, data analysis method(s)). More, the authors should present in detail the adaptation of the measurement instrument. I don’t understand how you choose the variables using relevant previous studies.

3.     Section 5. Conclusions- I think that some more words should be spent for commenting the obtained results and de conclusions of the paper (especially future research and the limitations of the research).

Author Response

Dear Reviewer:

Thank you for your comments concerning our manuscript entitled “The Impact of Clothing E-Store Image on Intention Based on Search and Purchase Phases: From the Perspective of Sustainable Marketing” (Sustainability-2065253). Those comments are all valuable and very helpful for revising and improving our paper, as well as the important guiding significance to our researches. We have studied comments carefully and have made correction which we hope meet with approval.  The main corrections in the paper and the responds to the reviewer’s comments are as flowing:

Point 1:  Section 1. Introduction- the authors should introduce the research questions and the structure of the paper here. More, the authors should introduce more valuable references, especially from MDPI journals.

Response 1: We are very sorry for our negligence of that. According to your suggestions, we have supplemented the three main research questions of this study in the introduction, and introduced the structure of the paper. At the same time, four valuable references from MDPI were introduced, which were published from 2018 to 2022 and were closely related to the research results of this study. These references are supplemented in the Introduction, Theoretical Background and Research Hypotheses.

Point 2: Section 4. Research methodology- this section is very briefly presented by the authors and also it concerns me the most. Please present the methodology of research (sampling design, data collection, data analysis method(s)). More, the authors should present in detail the adaptation of the measurement instrument. I don’t understand how you choose the variables using relevant previous studies.

Response 2: Considering the Reviewer’s suggestion, we rewrote Section 4 and renamed it Materials and Methods. This part includes four aspects: Questionnaire Design, Sampling Procedure, Ethical Considerations and Data Analysis. We introduced the design idea and process of the questionnaire in detail, supplemented the measurement scale, introduced the process of data collection and analysis methods in detail, and supplemented the consideration of ethical issues.

Point 3: Section 5. Conclusions- I think that some more words should be spent for commenting the obtained results and de conclusions of the paper (especially future research and the limitations of the research).

Response 3: We are very sorry for our negligence of that. In consideration of your suggestion, we split the original Discussion in Section 5. into two parts: Section 5. Discussion and Section 6. Conclusions, Implications, Limitations and Future Research. We have updated the application value of the paper, supplemented the limitations of the paper and assumptions for future research. After supplementation, the evaluation of this study is more practical and the exploration direction of future research is more clear.

Author Response

Dear Reviewer:

Thank you for your comments concerning our manuscript entitled “The Impact of Clothing E-Store Image on Intention Based on Search and Purchase Phases: From the Perspective of Sustainable Marketing” (Sustainability-2065253). Those comments are all valuable and very helpful for revising and improving our paper, as well as the important guiding significance to our researches. We have studied comments carefully and have made correction which we hope meet with approval.  The main corrections in the paper and the responds to the reviewer’s comments are as flowing:

Point 1: Introduction section should be more clarified. In this section, purpose of study is somewhat poor and it seems that this section is difficult to make readers understand why this research should be needed. Please provide how this study contribute to E-store industry more precisely.

Response 1: We are very sorry for our negligence of that. In consideration of your suggestion, we have revised the Introduction section. We added three main research questions, which further clarified the value of these questions. We mentioned in the revised version: A good image of e-stores is a beautiful picture in the minds of consumers. It not only symbolizes the strength of online retailers, but also attracts consumers to constantly browse and patronize online stores. With the development of online shopping, the competition between online stores is becoming increasingly fierce. As one of the antecedents that affect consumers' online search and purchase, how to identify and play the online store impression effect has been widely concerned by marketing scholars and online retail business operators. Therefore, it is of practical significance to establish the relationship model of e-store image, consumer attitude and consumer intention, reveal the mechanism of the impact of e-store image on consumer attitude and intention in the two stages of search and purchase, and propose corresponding sustainable marketing strategies for online retail enterprises according to the verification results.

Point 2: In this study, in my opinion, E-store image is one of the most important variable which include six dimensions. However,it seems that the process which these six dimensions are extracted from are not very clear. Please try to address how to provide six dimensions to explain e-store image. This process should be more clarified. In addition, it would be better if the authors account for each dimensions more specifically. Then, hypotheses may seem to be more stubborn.

Response 2: It is really true as Reviewer suggested that E-store image is one of the most important variable which include six dimensions. We are very sorry for our negligence of the description of the process of extracting these six dimensions. This part is supplemented in the Theoretical Background of E-Store Image. In Line 182-208, each dimension is defined and explained as well.

Point 3: According to demographic information, some specificcategory seems very radical such as female (gender), 19-24 (age), below 2000 yuan(monthly income), and junior college and undergraduate (education). This kind of pattern (the structure of participants in this study) may decrease reliability and validity which could negatively affect study. Therefore, the authors must convince your readers regarding this issue. For this issue, I would like to suggest that analyze the data again with data including  only  young  generation  participants  to  examine  e-store  image  perceived  from  young generation. It would also show interesting results. Please take into consideration. (In my opinion, this comment is the most critical in this study)

Response 3: In consideration of your suggestion, we have rewritten the demographic information section. The unnecessary classification is removed, and only important classifications such as age and education level are retained. 77.8% of the participants are young people aged 19 to 24 years, mostly college students. Contemporary college students have grown up in a more mature Internet era and become the first generation of aborigines of the Internet. The data analysis results of this study also reflect the online store impression of the young generation. 

Point 4: Line 66-67, unclear. TRA does not describe the background of e-store. I mean that if any prior studies examined TRA applied e-store as a background, please provide the studies more specifically.

Response 4: It is really true that TRA does not describe the background of e-store. In consideration of your suggestions, we have supplemented the research literature on the application of TRA in the field of e-store image. We mentioned in Line 76-82 of the revised version: Many scholars have used the theory of reasoned action (TRA) to study the attributes and images of e-stores. It is also found that consumers have formed different e-store attitudes and buying behaviors after perceiving different e-store images [5-8,15]. Their research shows that consumers can get an image by judging the attributes of online stores, thus deciding whether to buy. In other words, e-store image is an important antecedent variable affecting consumer attitudes [9], so we can construct the relationship of "E-store Image-Consumer Attitude-Online Shopping Behavior". 

Point 5: In  the  managerial  implication  section,  the  authors  tried  to  suggest  many  ideas  based  on  the result of this study. However, some of them are already applied by current e-store practitioners and they  are  very  common  strategies  in  real  industry  (such  as  design  of  the  store  atmosphere environment).  Therefore,  I  strongly  suggest  that  this  section  needs  more  creative  and  newly developed ideas to provide meaningful contribution toe-store industry.

Response 5: We are very sorry for our negligence of that. We split the original Discussion in Section 5. into two parts: Section 5. Discussion and Section 6. Conclusions, Implications, Limitations and Future Research. We have supplemented more creative and newly developed ideas in Section 6. For example, E-store operators should set up professional operation and art teams of stores, pay more attention to the design of the store atmosphere environment, such as forming a consistent overall style, color matching and aesthetic enjoyment, and updating the store layout with festivals or promotional activities to create an atmosphere, so that consumers can enjoy the beauty of searching or browsing in the store, promote the search behavior, and then enhance the consumers to purchase. Besides, Online clothing stores can provide consumers with good live broadcast scenes, live broadcast effects and clothing product experience. Under the live broadcast with good picture quality, customers can quickly integrate into the shopping scene, so that consumers can feel the satisfaction and happiness of online live broadcast shopping. Innovate and enrich the way and content of interaction in the live broadcast process, provide consumers with novel, interesting and immersive experience, and enable consumers to obtain good emotional experience, content experience and interactive experience in the live broadcast process. 

Round 2

Reviewer 1 Report

I'm satisfied with this new version of the manuscript. Perhaps the authors should have added more bibliographic references related to sustainability from MDPI group journals (for example a) Attar, R.W.; Almusharraf, A.; Alfawaz, A.; Hajli, N. New Trends in E-Commerce Research: Linking Social Commerce and Sharing Commerce: A Systematic Literature Review. Sustainability 202214, 16024. https://doi.org/10.3390/su142316024; b) Ceptureanu, S.I.; Ceptureanu, E.G.; Popescu, D.; Anca Orzan, O. Eco-innovation Capability and Sustainability Driven Innovation Practices in Romanian SMEs. Sustainability 202012, 7106. https://doi.org/10.3390/su12177106; c) Peña-García, N.; van der Woude, D.; Rodríguez-Orejuela, A. Recommend or Not: Is Generation the Key? A Perspective from the SOR Paradigm for Online Stores in Colombia. Sustainability 202214, 16104. https://doi.org/10.3390/su142316104).

Author Response

Dear Reviewer:

Thank you for your comments concerning our manuscript entitled “The Impact of Clothing E-Store Image on Intention Based on Search and Purchase Phases: From the Perspective of Sustainable Marketing” (Sustainability-2065253). Those comments are all valuable and very helpful for revising and improving our paper, as well as the important guiding significance to our researches. We have studied comments carefully and have made correction which we hope meet with approval.  The main corrections in the paper and the responds to the reviewer’s comments are as flowing:

Point 1: 1. I'm satisfied with this new version of the manuscript. Perhaps the authors should have added more bibliographic references related to sustainability from MDPI group journals (for example a) Attar, R.W.; Almusharraf, A.; Alfawaz, A.; Hajli, N. New Trends in E-Commerce Research: Linking Social Commerce and Sharing Commerce: A Systematic Literature Review. Sustainability 2022, 14, 16024. https://doi.org/10.3390/su142316024; b) Ceptureanu, S.I.; Ceptureanu, E.G.; Popescu, D.; Anca Orzan, O. Eco-innovation Capability and Sustainability Driven Innovation Practices in Romanian SMEs. Sustainability 2020, 12, 7106. https://doi.org/10.3390/su12177106; c) Peña-García, N.; van der Woude, D.; Rodríguez-Orejuela, A. Recommend or Not: Is Generation the Key? A Perspective from the SOR Paradigm for Online Stores in Colombia. Sustainability 2022, 14, 16104. https://doi.org/10.3390/su142316104).

Response 1: Thank you for your encouragement. According to your suggestions, we have supplemented these articles you mentioned. These documents are very helpful for raising questions, theoretical basis, assumptions and conclusions of this paper.

Reviewer 2 Report

I think most of my prior comments were revised. However, one very critical issue still remains.

1. In my first review comment, for the third point, I think you misunderstood what I meant. I did not mean you need to get rid of demographic information table. Rather, I meant the authors should account for how to generalize result from radical data that this study collected. To be specific, this study is more likely to account for the response of female (80%), young people (19-24) (78%), people whose income is below 2000 yuan and college student (87%). These explain greater than 70% of each group. Therefore, I concern that this Radical data structure may not be able to generalize the real world. In this study, how can you explain attitude of people who are older than 20s, male, people whose income is over 2000 yuan? (Again, this is very critical issue)

2.(minor) For the measures of model fit and reliability, please indicate which previous study cited the criteria.

Author Response

Dear Reviewer:

Thank you for your comments concerning our manuscript entitled “The Impact of Clothing E-Store Image on Intention Based on Search and Purchase Phases: From the Perspective of Sustainable Marketing” (Sustainability-2065253). Those comments are all valuable and very helpful for revising and improving our paper, as well as the important guiding significance to our researches. We have studied comments carefully and have made correction which we hope meet with approval.  The main corrections in the paper and the responds to the reviewer’s comments are as flowing:

Point 1: 1. In my first review comment, for the third point, I think you misunderstood what I meant. I did not mean you need to get rid of demographic information table. Rather, I meant the authors should account for how to generalize result from radical data that this study collected. To be specific, this study is more likely to account for the response of female (80%), young people (19-24) (78%), people whose income is below 2000 yuan and college student (87%). These explain greater than 70% of each group. Therefore, I concern that this Radical data structure may not be able to generalize the real world. In this study, how can you explain attitude of people who are older than 20s, male, people whose income is over 2000 yuan? (Again, this is very critical issue)

Response 1: We are sorry that we did not fully understand the first round of amendments. Thank you very much for your detailed explanation and giving us another opportunity to revise. Following your suggestion, considering that most of the respondents are young people aged 19 to 24, we took this part of the sample separately and re analyzed the data in this paper. And the results are reflected in the section 5. At the same time, the proposition of Introduction, hypothesis, conclusion and sustainable marketing strategy have also been greatly modified from the perspective of young people. All modifications are recorded with revision marks. The expenditure on clothes is the main part of young consumers' online shopping expenditure. Young people usually walk in the front of fashion and have high requirements for clothing consumption and updating. It is more meaningful to study their online shopping behavior.

Point 2: For the measures of model fit and reliability, please indicate which previous study cited the criteria.

Response 2: Sorry we ignored that. Considering your suggestions, we supplemented the literature on model fitting and validity test criteria.

Round 3

Reviewer 1 Report

I'm satisfied with revised form of the manuscript and I recommend the journal to publish the paper as it is.

Reviewer 2 Report

Thank you for your efforts. Now, I think this paper is ready to publish. 

Before you submit, please check it for typos again.